# Bridge Healing: A Pilot Project of a New Model to Prevent Repeat “Social Admit” Visits to the Emergency Department and Help Break the Cycle of Homelessness in Canada

**DOI:** 10.3390/ijerph20196845

**Published:** 2023-09-27

**Authors:** Matthew Robrigado, Igor Zorić, David A. Sleet, Louis Hugo Francescutti

**Affiliations:** 1Faculty of Medicine and Dentistry, University of Alberta, Edmonton, AB T6G 1C9, Canada; zoric@ualberta.ca; 2Rollins School of Public Health, Emory University, Atlanta, GA 30322, USA; davidasleet@gmail.com; 3Bizzell US, New Carrollton, MD 20785, USA

**Keywords:** homeless(ness), bridge healing, housing first, transitional housing, homeless patients, emergency department, The Eden alternative™

## Abstract

Homelessness continues to be a pervasive public health problem throughout Canada. Hospital Emergency Departments (EDs) and inpatient wards have become a source of temporary care and shelter for homeless patients. Upon leaving the hospital, homeless patients are not more equipped than before to find permanent housing. The Bridge Healing program in Edmonton, Alberta, has emerged as a novel approach to addressing homelessness by providing transitional housing for those relying on repeated visits to the ED. This paper describes the three essential components to the Bridge Healing model: partnership between the ED and a Housing First community organization; facility design based on The Eden Alternative™ principles; and grassroots community funding. This paper, in conjunction with the current pilot project of the Bridge Healing facilities, serves as a proof of concept for the model and can inform transitional housing approaches in other communities.

## 1. Introduction

Homelessness is a significant and pervasive issue in Canada that greatly impacts the health, security, and wellbeing of affected individuals. According to a report by Gaetz, Dej, and Richter (2016) [1], an estimated 235,000 people experience homelessness each year in Canada. This problem is persistent, with many individuals facing multiple and complex challenges that can lead to poor health outcomes. Homelessness can increase the risk of mental and physical illnesses, substance abuse, and mortality [2,3]. It can also lead to a lack of access to primary healthcare services. Homeless individuals often face barriers to accessing healthcare, including a lack of transportation, limited financial resources, and stigma [4]. The lack of access to primary care can result in unaddressed health needs, exacerbating pre-existing conditions and leading to further health problems [3,5].

The current response to homelessness in Canada relies heavily on emergency departments (EDs) acting as temporary sources of shelter, physical care, and food [6]. The admission of homeless patients into the hospital, known as a “social admit”, occurs when patients lack a reliable form of housing or transportation to reach outpatient care settings for follow-up appointments or simply need to be sheltered from weather events. Unfortunately, EDs are ill equipped to provide appropriate care to homeless individuals, especially those with chronic mental or physical illnesses that require continuous care. These conditions are common among the homeless population, making it difficult for EDs to address patient healthcare needs effectively [6,7]. Furthermore, this practice has the potential to worsen the conditions of homeless individuals as they are made vulnerable to hospital-acquired infections and experience a loss of privacy and freedom of movement upon being admitted as a hospital inpatient.

Upon discharge from the hospital, homeless individuals are no more equipped to acquire permanent housing than before visiting the ED. Moving from temporary shelters or acquaintances’ homes to EDs and back provides just enough food and warmth to survive but not enough for individuals to escape homelessness. As a result, a cycle of repeatedly visiting an ED for temporary care is perpetuated, leading to poor health outcomes and substantial costs to the healthcare system [5]. Anecdotally, the Royal Alexandra Hospital Emergency Department (RAH ED) in Edmonton (Alberta) reported seeing a homeless patient who had 556 documented visits to EDs in the region.

Recent surveillance data reveals that Edmonton EDs observed a 9.8% increase in visits by unique homeless people from 2019 to 2021, and the total number of visits by homeless people in the same period has increased by 10.5% [8]. These trends are at odds with the Alberta government’s recently publicized Healthcare Action Plan, which outlines immediate goals such as decreasing ED wait times and improving Emergency Medical Service (EMS) efficiency [9].

Although the Canadian Institute for Health Information (CIHI) requires hospitals to track homelessness in patients’ charts, current data likely underestimate the number of homeless patient visits because frontline ED workers are not required to ask about housing status [10,11]. Furthermore, there is no current standardized approach to assess homelessness in EDs, and the provision of a friend’s or shelter’s address often confounds reporting [11]. According to the CIHI, the average cost per acute care hospitalization day in Canada was CAD 7752 in 2018–2019. The cost varies by province or territory and by specific hospital. According to the 2021–2022 annual report from Alberta Health Services, the estimated cost to the healthcare system for a standard inpatient stay is over CAD 9000 [12].

As such, there is a need for a comprehensive approach to address homelessness in Canada, one that focuses on addressing the underlying causes of homelessness while improving access to healthcare at a lower cost. Supportive housing has been shown to be an effective solution for addressing homelessness and improving health outcomes [13,14]. Additionally, programs that address substance abuse, mental health, and employment support can also be beneficial for homeless individuals [2].

In acknowledging this cycle of homeless individuals visiting the ED and being discharged with minimal outside support, the Bridge Healing program was championed in Edmonton to provide an alternative housing and healthcare solution [15]. Bridge Healing is a model that figuratively bridges EDs with transitional housing facilities designed to provide wraparound healthcare, connect residents to community resources, and facilitate the acquisition of permanent housing. The program has the capacity to act as more than a temporary shelter by providing residents with opportunities to reintegrate back into the community through employment and social connection and prevent repeat visits to the ED for shelter and health care. The operating costs associated with housing and providing services for one resident per day in a Bridge Healing facility is approximately CAD 80, compared with thousands per day as a hospital inpatient. Residents are not charged during their stay at Bridge Healing. Hopefully, by equipping frontline ED staff with the ability to refer people to such facilities, the cycle of homelessness together with its demands and costs to the healthcare system can be broken.

To our knowledge, EDs in Alberta or in the rest of Canada have not been directly connected to a transitional housing program such as Bridge Healing. The model ultimately serves a dual purpose in substantially improving the long-term health outcomes of homeless people in Edmonton and lessening the capacity strain and wait times experienced in EDs. Since its inception in 2019, Bridge Healing has been endorsed by the Government of Alberta, Alberta Health Services, the Alberta Medical Association’s section of emergency medicine, the Royal Alexandra Hospital Foundation, and Edmonton Police Service. Bridge Healing’s media presence has been facilitated by major news outlets, including *The Globe and Mail*, Canadian Broadcast Corporation (CBC), and local and provincial divisions of CTV and Global News. Financial support for the project has come from grassroots community-level advocacy and from Lions Clubs International, the Edmonton Oilers Community Foundation, private donors, the City of Edmonton, and Alberta Health Services.

Bridge Healing launched a pilot project in January 2023, where three buildings—each comprising 12 residential units—became available for homeless patient referrals from the RAH ED. Within the first 6 months of its operation, Bridge Healing facilities have been accepting residents, 25% of which have been able to find permanent housing at this point. With this initial report, we describe the three essential components of Bridge Healing, how the project developed into its current state, and provide proof of concept of the model. Following the initial launch of Bridge Healing facilities, the program has received funding from the College of Physicians and Surgeons of Alberta (CPSA) for a longitudinal evaluation of the program’s efficacy. After the operation and assessment of this pilot project, if the program is found to be effective, we intend to facilitate the adoption of the Bridge Healing model in other communities to ultimately reduce the impact of homelessness across Canada.

## 2. Components of the Bridge Healing Model

### 2.1. Partnership between a City Emergency Department and a Community Health and Housing Center to Utilize a Housing First Approach

The operation of the Bridge Healing facilities and provision of wraparound healthcare services is delivered by Jasper Place Wellness Center (JPWC), a registered nonprofit organization that focuses on community building through affordable housing under the Housing First model [6]. JPWC has significant experience in facilitating community-based solutions to homelessness and has been an established community organization in Edmonton since 2011, having opened the city’s first low-intensity supportive housing facility.

The Housing First model prioritizes safe permanent housing as a necessary requirement for health and a starting point for developing healthy personal practices. Unlike traditional “treatment first” shelter approaches, which require potential residents to stop using drugs or alcohol before accessing the residence, Housing First models do not have such requirements. Instead, they aim to reduce harm by creating a safe environment that reduces opportunities for needle sharing and increases the likelihood of accessing treatment for drug-related medical concerns, should residents require it [16].

The increased adoption of Housing First models by community health centers such as JPWC has been correlated with an increased use of substance abuse treatment and subsequent quality of life improvements [17,18]. As such, Bridge Healing facilities focus on a harm-reductive approach, which contrasts with many traditional high-occupancy shelters that require clients to completely abstain from drugs or alcohol while using shelter services.

Residents of Bridge Healing facilities can stay for an average of 30 days (depending on their individual needs) and have access to wraparound services provided by JPWC, including mental health support, addictions and employment counseling, access to outpatient medical services through the JPWC Primary Care Clinic, food security, and connections to permanent supportive housing. Trained supportive housing staff from JPWC are always present on site to support residents, helping to support the development of a homeless person’s ability to live independently.

The direct partnership between the RAH ED and JPWC is a novel collaboration that connects an acute care center and a community health and housing center. It provides a streamlined process for moving homeless patients from the hospital to transitional housing. This partnership is a model for other communities to consider where homeless populations are not receiving adequate care in their continuum of care. Community organizations such as JPWC can play a critical role in filling gaps where medical services fall short. This collaboration has the potential to improve the lives of homeless individuals and reduce the burden on hospital systems. As a result, other acute care centers and community health organizations should consider this model when designing programs to address the needs of homeless populations. By working together, we can create sustainable and effective solutions to address the complex challenges faced by homeless individuals in our communities.

### 2.2. Referral Process

As per the current ED triage protocol, all patients presenting to the RAH ED will be screened for their housing status. Patients reporting as houseless will be offered a referral by a social worker to discuss their eligibility for referral to Bridge Healing based on the following criteria:A.18 years of age or older.B.Self-identified as being without housing and willing to reside individually.C.Able to safely reside in a communal building.D.Indicates a desire to actively work towards finding permanent housing.E.No active medical, surgical, or psychiatric concerns that warrant hospital admission (i.e., the patient is being discharged from the ED).F.Ability to use the restroom, manage medications, and transfer (ambulate) independently.

If the above inclusion criteria are met, the patient will be discharged from the ED to Bridge Healing by taxi. Following patient evaluation for eligibility, the social work team will document their individual outcome onto ConnectCare (the Alberta provincial clinical information system) to continually monitor referral eligibility and records of outcomes.

### 2.3. Facility Design Utilizing The Eden Alternative^TM^

Transitional housing facilities designed with The Eden Alternative™ concept aim to address prevalent concerns of loneliness, helplessness, and boredom, which can negatively impact resident wellness. The Eden Alternative™ is a design philosophy developed in response to these concerns, originally to improve resident wellbeing in nursing homes in the United States as occupancy rates were increasing rapidly [19]. The aim of The Eden Alternative™ is to create living spaces that naturally facilitate companionship and mutually beneficial community building through criteria including low occupancy and a particular focus on shared spaces. Therefore, residents can feel socially supported while receiving care while simultaneously making positive social impacts on others [19]. The adoption of The Eden Alternative™ in long-term care homes has been shown to increase resident quality of life, satisfaction with services, and staff retention and job satisfaction rates [20]. An American study that qualitatively evaluated the impact of implementing The Eden Alternative™ design philosophy into the operation of nursing homes found stronger relationships between residents and staff as well as a sense of staff empowerment during postintervention interviews [21]. A similar 5-year case-control study also in the United States found similar results and a notable reduction in resident admission to hospital and depressive symptoms [22]. The Bridge Healing program has thus adopted this evidence-based approach to encourage interpersonal relationships and feelings of belonging between residents.

Early in the development of the Bridge Healing model, a strong sense of community within the facilities was identified as essential to the project after evaluating successful Tiny Village approaches to homelessness worldwide [15]. The most notable design elements of each building in Bridge Healing are the limit of 12 resident units (i.e., 6 units on the second floor, 6 units on the third floor and the first floor reserved entirely for community space), which is intended to aid in naturally building a sense of community within the facilities (see Figure 1). Limiting occupancy to 12 people per building creates opportunities for connection building and accountability between residents. In contrast, high-occupancy shelters can lead residents to feel as though they have been “institutionalized”, and they are less likely to get to know each other. Relationships between residents are facilitated by the communal pantry services, laundry room, and staff present on the main floor. Notably, there is no secluded area for the staff in the facility, ensuring that they are incorporated into the building’s community and limiting the potential for social barriers between staff and residents to form.

To promote physical activity and engagement, each housing facility’s outdoor area contains a gardening space available for use by residents. Communal gardening areas have been used successfully in other facilities adopting The Eden Alternative™ to introduce novelty, reduce helplessness, and improve residents’ physical condition [20].

Resident security is also a crucial factor in creating a sense of community and belonging. Keeping resident units off the street level reduces the chance of their behavior being watched or monitored by outsiders who may pose risks to their health. Uninvited guests are also unable to interact with residents as each unit has visual access into the foyer through an intercom. While the pilot project and ongoing assessments will inform practical areas of improvement for the facility, it is noteworthy for other community organizations and transitional housing operators to consider designs that facilitate these community-building assets for single adults experiencing a lack of housing and social support.

### 2.4. Formal Evaluation of the Pilot Project

After the program’s launch in January 2023, Bridge Healing has acquired funding from the CPSA ‘Healthier Albertan Grant’ to execute a formal longitudinal evaluation starting in June 2023. The primary objective of the evaluation is to quantitatively determine whether patients without housing presenting to the RAH ED are able to transition to Bridge Healing housing. Secondarily, data will be collected based on if patients are still residing in Bridge Healing at 7 days and 30 days after their initial intake. Within-person comparisons will be performed to observe any change in an individual’s number of ED visits, inpatient length of stays, and number of EMS transports at time points 3 months before their initial ED visit and 3 months after leaving Bridge Healing. Patients eligible for housing with Bridge Healing will be presented with the opportunity to participate in the program evaluation prior to discharge from the ED. If consent is provided, individuals will complete a baseline survey to collect data on their past experience and length of houselessness, currently reported quality of life, and other important covariates such as health status. Prior to leaving Bridge Healing, individuals will be asked to complete a follow-up survey to re-evaluate their quality of life and covariates, as well as to provide feedback about their experience in the Bridge Healing program to inform continuous program refinement.

### 2.5. A Grassroots Community Initiative with Funding That Has Minimal Operational Restrictions

While supportive housing has been identified as a critical element for addressing homelessness and improving the health of individuals experiencing homelessness [15], funding is often a critical element of success. Securing funding without associated operational restrictions can be challenging for many supportive housing projects. To successfully obtain funding for the Bridge Healing project, its champions heavily advocated for grassroots community involvement from the project’s early phases (Figure 2).

The concept and potential funding for Bridge Healing was initiated when a local emergency physician at the Royal Alexandra Hospital (RAH) ED recruited a business developer at a health analytics conference at the Northern Alberta Institute of Technology (NAIT) in 2018. The staff physician had previously worked with the business developer as a client in 2000, and with their existing relationship, they were able to recruit a retired project manager to develop the Bridge Healing project team. Graduate students at NAIT, who were mentored by the project manager, developed early iterations of the Bridge Healing model for capstone projects (a capstone project is a culminating academic project that synthesizes and applies the knowledge and skills acquired throughout a degree program to address a real-world problem or issue [23]) as a mobile application that could connect those leaving the ED with immediate housing.

After developing the Bridge Healing idea and integrating social workers at the RAH ED, the project team began to explore community partnerships and funding opportunities. This search was largely driven by continued advocacy for the project through professional connections. Private endorsers and funders were among those recruited to help support and finance Bridge Healing along with other residents of the neighborhood where the facilities are now located. The involvement of Lions Clubs International, where the business developer had a strong personal connection, was instrumental in funding the facilities.

The establishment of Bridge Healing, up to the point of being recognized by the RAH Foundation, the City of Edmonton, and Alberta Health Services, was made possible through the direct contact and advocacy of the project team for the healthcare needs of the homeless population. The project team’s ongoing efforts to secure funding and support from various sources were instrumental in the RAH Foundation’s commitment to supporting Bridge Healing.

Grassroots initiatives such as Bridge Healing have been shown to be effective in addressing issues related to homelessness and supportive housing [15,24]. Community involvement, advocacy, and secure funding were critical to the success of the Bridge Healing program in Edmonton. Strong personal and professional connections were important in gaining endorsements and developing partnerships among local organizations and individuals. In addition, capstone projects such as the ones developed by NAIT graduate students can provide additional support and training experience for emerging leaders in public health and medicine. The Bridge Healing program demonstrated that by leveraging the knowledge and skills of a diverse group of stakeholders, including healthcare professionals, business developers, and community members, grassroots efforts can be made to develop alternative approaches to ongoing population-level concerns such as homelessness.

## 3. Conclusions

The Bridge Healing program is a new model that aims to break the current cycle of homelessness and prevent unsustainable, repeated “social admits” to emergency departments in Edmonton, Alberta, Canada. Existing as a collaboration between healthcare providers, community organizations, and government agencies, the Bridge Healing model is, to our knowledge, the only housing program that links these areas of public services together. It is designed to provide healthcare services in a more appropriate setting than EDs and act as transitional housing for homeless individuals who would not normally receive direction to permanent housing options had they been a “social admit”. This design facilitates Bridge Healing’s ultimate goal of reintegrating residents into the community and preventing poor health outcomes as well as unnecessary costs to the healthcare system. The model has been endorsed by multiple stakeholders and is financially supported by grassroots community-level organizations. Its success will be determined by a formal evaluation of the pilot project sponsored by the College of Physicians and Surgeons of Alberta. We hope that by presenting this program’s approach we can encourage others in different cities to consider solutions to repeated ED visits through housing.

## Figures and Tables

**Figure 1 ijerph-20-06845-f001:**
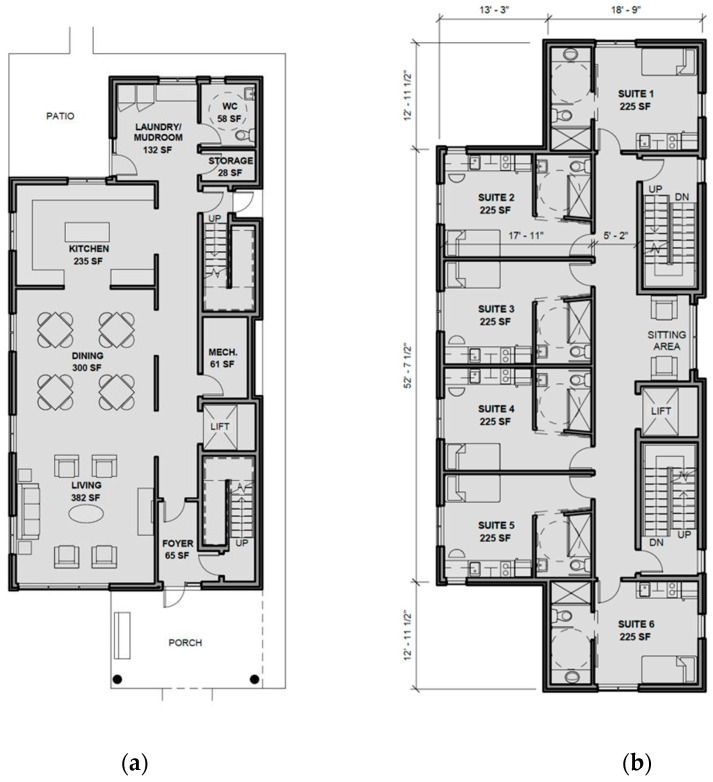
Bridge Healing facility floorplan with consideration of The Eden Alternative^TM^. (**a**) First floor. (**b**) Second floor.

**Figure 2 ijerph-20-06845-f002:**
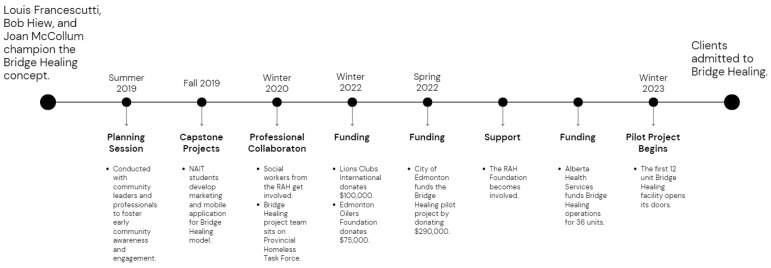
Bridge Healing development timeline.

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
