# Peer review of "Bridge Healing: A Pilot Project of a New Model to Prevent Repeat “Social Admit” Visits to the Emergency Department and Help Break the Cycle of Homelessness in Canada"

_ijerph, 2023, doi:10.3390/ijerph20196845_

Round 1

Author Response

  1. The largest difference between Bridge Healing and traditional high occupancy shelters is the design based on The Eden Alternative. As described in section 2.3 and shown in Fig. 1, the facilities are designed to facilitate community building through communal areas and build rapport amongst residents by maintaining a maximum occupancy per building at 12 persons.
  2. The conclusion section is rewritten, and superfluous adjectives were removed.
  3. Addressing feedback points 3 & 4, the conclusion was rewritten and shortened overall to avoid redundancy with points already made.

Thank you very much for the feedback.

Reviewer 2 Report

Congratulations on a well-written manuscript about a novel practice innovation. I have a few minor comments - mostly to satisfy my own curiosity and potentially that of other readers of the article. 

Can you include reference to how this initiative will be evaluated? This has the potential to grow nationally and internationally, but will rely on funding. To that end, data will be required to convince funders of the benefits and to share learnings with other contexts. 

Since this journal will reach an international audience, I suggest you indicate where relevant how this model can have international relevance - especially in contexts where both the Housing First and Eden Alternative concepts are / should be considered.

There is no mention of the referral process between the ED and the BH programme. Since you have highlighted the challenge that information about homelessness in the ED is fraught, it seems an important point in the process to ensure that people are identified early (before 500 visits!). 

I am keen to learn more how the wraparound service is provided in units with only 12 people. This can potentially be resource intense. It is not the point of this article to report on this in detail (maybe there is another manuscript due on the business and operational model - with evaluation data?), but I think a mention of this will be worthwhile. 

Author Response

  1. Evaluation methods for the efficacy of Bridge Healing to be conducted this coming spring-summer were included. In short, the program has acquired funding from the provincial college of physicians to conduct a longitudinal study on the program.
  2. Section 2.3 paragraph 1 has been revised to include specific examples of American studies that evaluated the implementation of The Eden Alternative. We hope these references can show that the design philosophy is evidence-based, which is why it was integral to the design of the Bridge Healing facilities.
  3. The addition of section 2.2 provides a description of the referral process for patients experiencing houselessness to be housed in Bridge Healing. We have also provided the currently used 'inclusion criteria' which is the basis for the intake evaluation by the social work team at the Royal Alexandria Hospital emergency department.
  4. Yes there is another manuscript planned to coincide with the upcoming longitudinal evaluation of the Bridge Healing program. We think with this manuscript covering the pilot project that there is value in presenting our approach so that readers can think about alternative solutions (such as Bridge Healing) to ongoing repeat ED visits and urban homelessness.

Thank you very much for your kind words and constructive feedback.

Reviewer 3 Report

1.   (Line 25) Comment: In reality, homelessness is not a health problem per se, it is a social problem and as such is the cause of multiple vulnerabilities in the affected population including health problems. Please reformulate this sentence

 2. The article proposes a description of the model, however does not present data on how this model works or even user perceptions of this facility.  The authors need to add data and results that verify the functionality in practice of the model they propose to solve the public health problem they pose.

3. The conclusions are poor and lack scientific support, because the arguments are not supported or based on information that can be collected in homeless communities and only mention disarticulated discussions of actors and experts, therefore this is a work that requires more data and more extensive research.

Author Response

  1. We agree with this point and have revised the sentence accordingly
  2. Longitudinal data collection, including qualitative experiences of patients, is currently underway. We have included a description of the evaluation process and objectives, while the reporting of findings from this evaluation are planned for a future manuscript.
  3. We agree that there is no evaluation data to provide in this manuscript. We defined the purpose of this manuscript to describe a 'proof of concept' for the process of establishing this program, and to encourage readers to think about alternative solutions (such as Bridge Healing) to apply to urban homelessness and repeat ED visits. As such, we have added to the title "A pilot project..." and put an emphasis into this manuscript on our evaluation objectives.

Thank you very much for taking the time to read our manuscript and provide feedback.

Reviewer 4 Report

Dear authors, 

Thank you for the opportunity to review your manuscript. You have described an innovative way to reduce repeated admissions to the Emergency Department (ED) and it appears that there is a lot of potential to improve homelessness people presenting at an ED. Your manuscript is well written, but the content is purely descriptive. 

In your description of the Bridge Healing Model you draw a strong conclusion (line 148-150) that your model should be adopted by other healthcare organisations. However, there is no evidence provided how effective your intervention is.  Another recommendation is made (line 193-194) for other community organizations to follow your model and this based on a pilot intervention without an evaluation in place. This does not make sense. Line 233 and 234 provides draws a conclusion that key stakeholder engagement makes a difference in the lives of homeless individuals, but what this based on?

It appears that you have described your intervention, which is a good introduction to your paper, but there is no method, findings section. I would like to suggest to consider resubmitting once the intervention has been evaluated. 

Author Response

Thank you very much for your time reviewing our manuscript and the constructive feedback you provided.

To address your comments, we have revised the strong language around recommending the adoption of our model (lines 148-150, 193-194, 233-234). We have also included a description of the longitudinal evaluation process that is currently in its initial stages (please see the added section 2.4). 

We plan for another manuscript to be made reporting the findings of the longitudinal evaluation of the Bridge Healing program. The purpose of this manuscript was defined as a 'proof of concept' for the establishment of an alternative transitional housing program, and to present readers with an opportunity to consider alternative solutions (such as Bridge Healing) to urban homelessness and repeat ED visits.

Round 2

Reviewer 3 Report

Thank you for incorporating your corrections. The manuscript already has the merit to be published.

Author Response

Thank you for taking the time to review our manuscript.

Reviewer 4 Report

Dear authors,

Thank you for addressing my comments.  The major issue of not having data  to support your model of care remains. Including initial evaluation data would strengthen your arguments that your model can reduce ED presentations.

Author Response

Hello,

Thank you for taking the time to review our manuscript and provide constructive feedback. We agree that including evaluation data of the pilot program would strengthen our argument that Bridge Healing is effective at increasing the acquisition of permanent housing and reducing "social admits" to hospitals.

In light of this, we have added section 2.5 - Preliminary Evaluation (lines 242 - 263), where we discuss key takeaways from an internal review of Bridge Healing 6 months after the program's launch. The section outlines both the positive effects of the program thus far and notably, actionable items for continued quality improvement.

We look forward to hearing your impression of this revised manuscript.